# How does inquisitiveness matter for generativity and happiness?

Junichi Hirose[1,2], Koji Kotani[2,3,4,5]*

1 Multidisciplinary Science Cluster, Collaborative Community Studies Unit, Kochi University, Kochi, Japan, 2 School of Economics and Management, Kochi University of Technology, Kochi, Japan, 3 Research Institute for Future Design, Kochi University of Technology, Kochi, Japan, 4 Urban Institute, Kyushu University, Fukuoka, Japan, 5 College of Business, Rikkyo University, Tokyo, Japan

* kojikotani757@gmail.com

**Data Availability Statement:** All relevant data are within the manuscript and its Supporting information files.

**Funding:** This work was supported by the Japan Society for the Promotion of Science as the Grant-in-Aid for Scientific Research B (19H01485) of KK.

## Abstract

Inquisitiveness (curiosity & acceptance to something and someone different) is the main engine for one person to initiate some relation, and the literature has established that maintaining nice relationships with friends, family and general others contributes to generativity and happiness. However, little is known about how generativity and happiness are characterized by inquisitiveness. We hypothesize that inquisitiveness is a fundamental determinant for generativity and happiness, empirically examining the relationships along with cognitive, noncognitive and sociodemographic factors. We conduct questionnaire surveys with 400 Japanese subjects, applying quantile regression and structural equation modeling to the data. First, the analysis identifies the importance of inquisitiveness in characterizing generativity in that people with high inquisitiveness tend to be generative. Second, people are identified to be happy as they have high generativity and inquisitiveness, demonstrating two influential roles of inquisitiveness as direct and indirect determinants through a mediator of generativity. Overall, the results suggest that inquisitiveness shall be a key element of people's happiness through intergenerational and intragenerational communications or relations.

## Introduction

Curiosity and acceptance are important elements for one person to gain creativity, fulfillment and views [1–4]. A child's tendency to ask a question shall be an initial step of building human relations and learning various things. In the literature, such a tendency is conceptualized as "inquisitiveness" representing curiosity & acceptance to something and someone different [5, 6]. Frazier et al. [7] examine adult-child conversational exchanges by focusing on young children's questions and adult's answers, claiming that such communications provide important bases for children's future life, especially regarding how they are able to grow through human interactions. Moreover, it is established that having and keeping nice relationships with family, friends and general others contribute to generativity and happiness [8–12]. Given this state of affairs, individual tendencies to be curious about and/or accept something and someone

The publication procedure was partly supported by the research fund of JH from Kochi University.

**Competing interests:** The authors have declared that no competing interests exist.

different (or new) may be a main engine for one person to be not only interactive with people in different generations but also happy. Therefore, this research addresses the role of inquisitiveness for generativity and happiness.

Erikson [13] introduces the concept of generativity and defines it as a concern regarding the establishing and guiding of future generations in the life-span theory of personality development. Generativity is expressed in bearing and raising children but is by no means limited to the domain of parenthood [8]. Various activities and behaviors concerning future generations for helping and teaching something useful and interesting to young generations, are also considered expressions of generativity [14, 15]. Some scales of generativity have been developed to quantify such people's activities, behaviors and concerns, e.g., the Loyola generativity scale (LGS) and the generative behavior checklist (GBC) [8, 10, 16, 17]. Utilizing these scales, previous studies have characterized generativity concerning psychological and sociodemographic factors, such as aging, education, gender, health, income, marital status, political view, type of societies and value orientation [11, 15, 18–22]. Overall, it is established that age, marital status and type of societies are main determinants of generativity.

Happiness is taken to be a term to represent an outcome of a "good life," where people act and behave to seek happiness [23–26]. In this paper, we interchangeably use the term "wellbeing" to refer to "happiness." Maslow proposes a theory based on psychological needs and gratification processes, suggesting that people are happy as they become wealthy, i.e., Maslow's hypothesis [27]. To examine this hypothesis, several researchers have developed and refined the measurements, such as the subjective happiness scale (SHS) and satisfaction with life scale (SWLS) [28–30]. Veenhoven [31] and Diener and Diener [32] examine the hypothesis with cross-country level data utilizing happiness scales and conclude that wealth can account for variation in happiness across countries to a certain extent; however, there should be some other important predictors. Following these works, the literature has focused on how happiness is associated with cultural, sociodemographic and personal factors, other than wealth or income, including education, gender, marital status, self-esteem, human relations, optimism and extraversion [33–40]. Overall, it is established that aging, income, human relationships, personality traits and value orientations matter for characterizing happiness [41–47].

Previous studies have examined the relationship between generativity and happiness, often along with social preferences [10, 15, 48–55]. Aknin et al. [49] conduct survey experiments with 51 students of the University of British Columbia, claiming that social preferences are positively associated with happiness and there exists a positive feedback loop between the two. Timilsina et al. [15] compare prosociality and generativity between rural and urban people by conducting survey experiments in Nepal. They find that rural people are more prosocial and generative than urban ones, and claim that prosocial orientation shall contribute to generativity. Building upon Timilsina et al. [15], Shahen et al. [55] conduct similar types of survey experiments in rural and urban areas of Bangladesh, collecting data on happiness and generativity along with prosociality and other variables. They establish that generativity is a robust and consistent predictor of happiness, controlling for prosociality and some other key sociodemographic factors in the analyses. Overall, these studies suggest that generativity and prosociality can influence happiness [9, 10, 56–58].

Inquisitiveness is a concept to represent curiosity & acceptance of something and someone different and/or new, and those with such inquisitiveness tend to start communications with others by asking questions [59–63]. After some development of the scales for inquisitiveness by Facione et al. [64], Hirayama and Kusumi [59] and Hogan and Hogan [65], some studies have been conducted to address how an inquisitive person behaves in terms of learning from and engaging with people regardless of their backgrounds, positions and roles as well as how such behaviors may lead to creative problem solving for nursing and schooling [61, 65–68].

Hirayama and Kusumi [59] conduct questionnaire surveys with 426 Japanese university students and analyze the critical thinking attitudes on the process of drawing a conclusion. They find that inquisitiveness is an essential factor in reaching a conclusion not bounded by people's beliefs. Nakagawa [69] also demonstrates that inquisitiveness is positively correlated with how people are well prepared for possible future disasters by conducting questionnaire surveys in Japan. Another group of studies analyze the role of inquisitiveness in leadership studies at schools and workplaces, generally confirming its importance in experiments and the fields [61, 63, 70, 71]. Overall, inquisitiveness is a powerful source of engines that increases the motivation and behaviors in some situations, triggering people's communications with others and their interactions with unfamiliar environments [60, 72–74].

No previous works have addressed how generativity and happiness are characterized by inquisitiveness, while both of these concepts are known to be highly concerned with how people build and keep relationships with family, friends and general others. Inquisitiveness is considered an important factor to trigger communications, being conjectured to contribute to maintaining nice human relations. Therefore, we hypothesize that inquisitiveness is an important determinant of happiness and generativity, empirically examining the relationships along with noncognitive, cognitive and sociodemographic factors in a single analytical framework. To this end, we conduct questionnaire surveys with 400 Japanese subjects to collect data, following previous studies that analyze the relationship between behaviors and happiness with cross-sectional data [75–79]. There are several studies that apply cross-sectional data analyses, such as mediation analysis and regressions, to examine the relationships among personality traits, behaviors and happiness [75, 76, 79]. With this data, our research addresses the following two open questions. (1) Does inquisitiveness play a role in generativity? (2) How does inquisitiveness, along with generativity, affect people's happiness?

## Materials and methods

### Participants and procedures

We conduct questionnaire surveys with 400 subjects sourced from the registered participant pool of a web-based survey research organization, Cross Marketing Inc., in Japan. The sample size is partly determined by the budget and time constraints facing us. Subjects' mean age is 47.79 years with a standard deviation = 16.74, ranging between 20 and 88 years. The survey area is divided into urban and nonurban ones according to a population density of 500 people $km^{-2}$. If the population density at the place where a subject lives is above the threshold, it is urban. Otherwise, it is nonurban. Literature establishes that prosociality differs between rural and urban areas in some developing countries [55, 80, 81]. Therefore, we take the samples from urban and nonurban areas, considering and controlling for such possibility in statistical analyses. This survey collects a sample of 200 subjects each in urban and nonurban areas (400 subjects in total) with information about (i) sociodemographic factors, such as age, gender, household income, marital status, educational background, family characteristics, (ii) generativity (a concern in guiding the next generation), (iii) subjective wellbeing (SWB) as happiness, (iv) inquisitiveness (curiosity & acceptance to something and someone different and/or new) and (v) social value orientation (as a proxy for social preferences). The variables we collect in this survey can be categorized into cognitive, noncognitive and sociodemographic factors in relation to SWB, as described in Fig 1.

### Measures

We employ the satisfaction with life scale (SWLS) to measure subjects' life satisfaction in our survey, wellbeing is a part of happiness [44]. The SWLS is an established measure of life

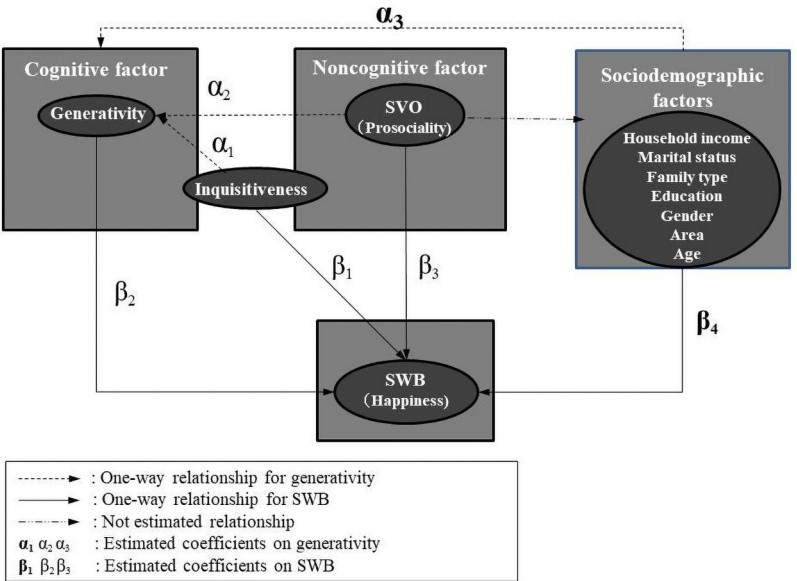

**Fig 1. A conceptual framework.** A conceptual framework describing the relationships concerning SWB among cognitive, noncognitive and sociodemographic factors.

satisfaction and is known as a concept that is central to the research area of subjective wellbeing (SWB) (see, e.g., [21, 28, 82, 83]). Validation is carried out across ages, countries and genders [28, 37, 84] and the components consist of several aspects (i.e., affective, intrinsic and extrinsic ones) [28, 85]. The affective aspect of life satisfaction refers to emotional elements, whereby levels of positive and negative ones are used to indicate the status of SWB [85]. In this case, the level of SWB is measured by psychological instruments, such as Ryff's psychological wellbeing scale [86]. The extrinsic aspect of life satisfaction refers to a relativistic judgment, whereby comparing oneself with others is used to indicate the status of SWB. In this case, the level of people's SWB is measured by instruments, such as the subjective happiness scale (SHS), as compared to that of their peers by stating "Compared to my peers, I consider myself," and its anchor is "less happy" and/or "more happy" [30].

This research focuses on intrinsic happiness, not limited to positive and negative emotions, employing the SWLS, which is designed to measure self-recognition of SWB [28, 84, 87]. The items of the SWLS include five short statements: (1) "In most ways, my life is close to my ideal," (2) "The conditions of my life are excellent," (3) "I am satisfied with my life," (4) "So far, I have gotten the important things I want in life" and (5) "If I could live my life over, I would change almost nothing." Each item scores on a 7-point Likert scale, ranging from 1 = "Strongly disagree" to 7 = "Strongly agree," and the total scale scores are the sum of the five-item scores, ranging between 5 and 35. The higher the scores are, the greater life satisfaction is. The scores are categorized as extremely satisfied ($31 \sim 35$), satisfied ($26 \sim 30$), slightly satisfied ($21 \sim 25$), neutral (20), slightly dissatisfied ($15 \sim 19$), dissatisfied ($10 \sim 14$) and extremely dissatisfied ($5 \sim 9$).

For generativity, researchers have developed several measurements to assess individual differences in consideration of its various aspects [11]. The Loyola generativity scale (LGS), which shall be considered a cognitive factor, is employed to measure "generative concern," as it is the most commonly used one in the literature (see, e.g., [8, 11, 18, 20, 22, 88–90]). The LGS scale contains a list of 20 questions, of which 6 questions are reverse questions. Another

popular scale for generativity is the generative behavior checklist (GBC) that scores on "generative behaviors" in the past two months [11, 16]. Both the LGS and GBC are established to display positive associations, demonstrating consistency between generative concerns and behaviors [16]. We decide to use the LGS rather than the GBC because we realize that some questions in the GBC shall be difficult for many Japanese people to answer because of the absence of such opportunities and experiences (e.g.,"Babysat for somebody else's children," "Taught Sunday school or provided similar religious instruction").

The items of the LGS include statements, such as (1) "I try to pass along the knowledge I have gained through my experiences," (2) "I have important skills that I try to teach others," (3) "I feel as though I have made a difference to many people," (4) "I have made and created things that have had an impact on other people," (5) "I have made many commitments to many different kinds of people, groups and activities in my life" and (6) "I do not volunteer to work for a charity." Here, question (6) is considered the reverse one. Subjects need to choose one of four options for each statement. "Zero," "one," "two" or "three" scores indicate how often the statement applies to subjects (Mark "zero" if a statement never applies, mark "three" if the statement applies very often or nearly always). In the case of reverse questions, we calculate the reverse score (i.e., zero becomes three, one becomes two, two becomes three and three becomes zero). The generativity score for each subject is computed as the sum of the scores for all 20 items. The theoretical range is between 0 and 60, being calculated as the sum of the scores from the LGS questions, and Cronbach's alpha for this measure is 0.90 in our sample.

We employ the inquisitiveness scale in our survey, which is a subscale of the critical thinking disposition scale developed by Hirayama and Kusumi [59]. This instrument is used to measure one's disposition for curiosity & acceptance of something and someone different and/or new [59, 69, 91]. This subscale consists of ten items, including (1) "I want to interact with people with various ways of thinking and learn a lot from them," (2) "I want to keep learning new things throughout my life," (3) "I like to challenge new things," (4) "I want to learn about various cultures," (5) "Learning how foreigners think is meaningful to me," (6) "I am interested in people who have a different way of thinking," (7) "I want to know more about any topic," (8) "I want to learn as much as possible, even if I do not know if it is useful," (9) "It is interesting to discuss with people who have different ideas than me" and (10) "I want to ask someone if I do not know." The items are rated from 1 = "Strongly disagree" to 5 = "Strongly agree." The theoretical range is between 10 and 50. This subscale is established as a reliable measure for influencing people's behaviors and attitudes in many important contexts, such as disaster management [69].

We use the SVO game with the "slider method" to identify subjects' social preferences as prosocial or proself [92]. Fig 2 shows the six items of the slider measure that gives numbers to represent outcomes for oneself and the other in a pair of people where the other is unknown to the subject. Subjects are asked to choose among the nine options for each item. Each subject chooses her allocation by marking a line that defines her most preferred distribution between herself and the other person. The mean allocation for herself $\overline{A}_s$ and that for the other person $\overline{A}_o$ are calculated from all six items (see Fig 2). Then, 50 is subtracted from $\overline{A}_s$, and $\overline{A}_o$ to shift the base of the resulting angle to the center of the circle (50, 50). The index of a subject's SVO is given by $\text{SVO} = \arctan\frac{(\overline{A}_o)-50}{(\overline{A}_s)-50}$. Depending on the values generated from the test, social preferences are categorized as follows: 1. altruist: SVO > 57.15˚, 2. prosocial: 22.45˚ < SVO < 57.15˚, 3. individualist: −12.04˚ < SVO < 22.45˚ and 4. competitive: SVO < −12.04˚.

The SVO framework assumes that people have different motivations and goals for evaluating resource allocations between themselves and others. Also, the SVOs are established to be stable for a long time (see, e.g., [93, 94]). Subjects that go through the six primary items in the

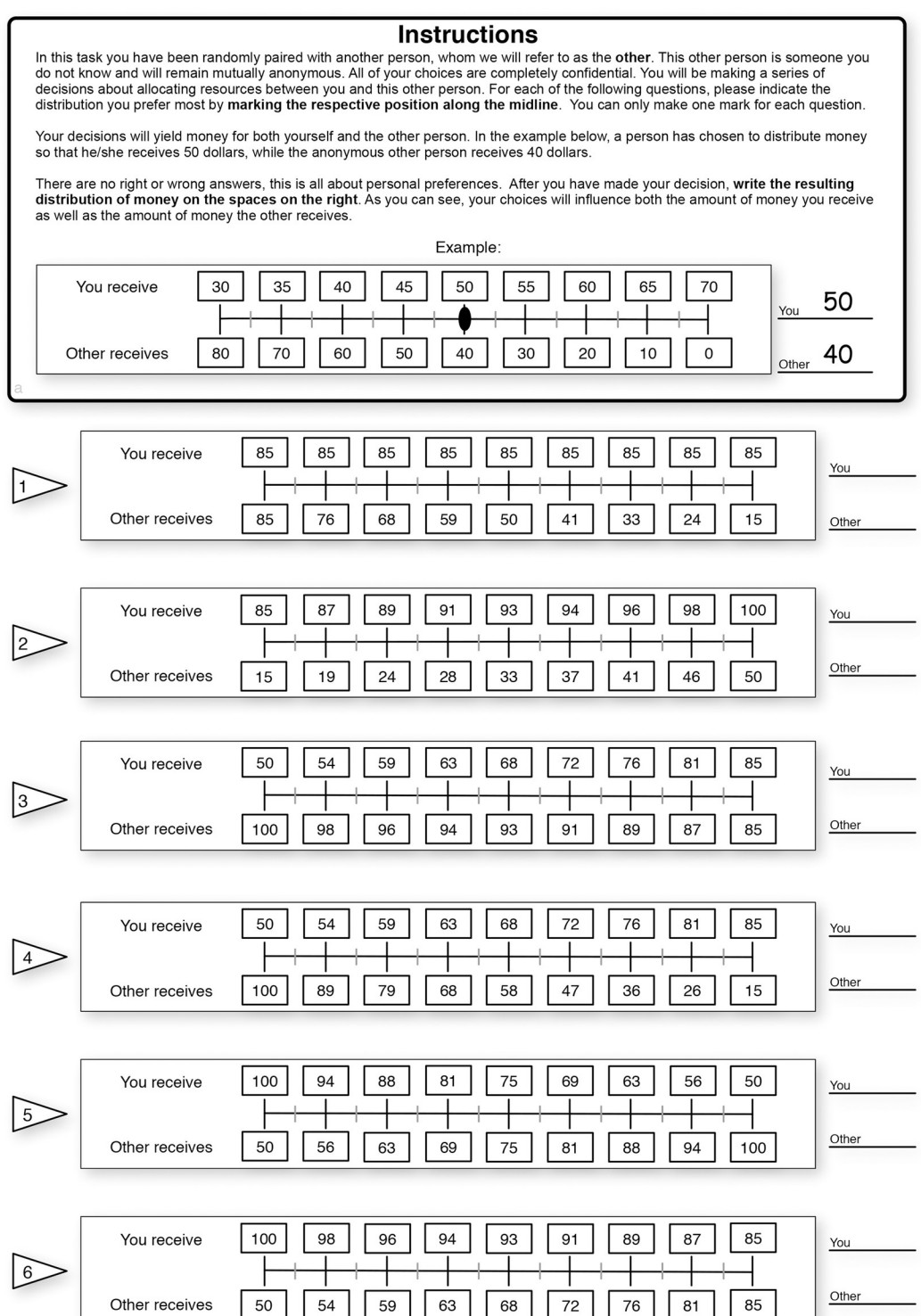

**Fig 2. Instructions of SVO.** Instructions of the "slider method" for measuring the social value orientation [92].

slider method are known to give complete categories of social preferences [92]. As has been done in the research of psychology, we simplify the four categories of social preferences into two categories of prosocial and proself types; "altruist" and "prosocial" types are categorized as prosocial subjects, while "individualist" and "competitive" types are categorized as "proself" subjects (see [92]). Subjects are informed that the units in this game are points, meaning that the more points they get, the more real money they will earn. For details, see the instructions in Fig 2. Our survey experiments are conducted with real monetary payments in the SVO game. This game is designed to motivate subjects to seriously perform in the survey experiment, considering their opportunity costs of time and their true revelation of social preferences.

One session takes 5 to 8 minutes. An exchange rate is applied to the points in the games to determine the monetary reward, and subjects receive a maximum of 150 JPY ($\approx$ 1.37 USD) and an average of 104 JPY ($\approx$ 0.95 USD) in the game. The decisions in this game are conducted in complete privacy. To compute the payoffs of subjects, we randomly match one subject with another to form a pair. The payoff for each subject in the game is the summation of the points from 6 selections by an individual, as "You," and 6 selections by the partner, as "Other." We explain the methods of random matching and payoff calculation with information on the exchange rate 1 point is converted to 1 JPY) for the real monetary incentive for subjects before starting the game. Subjects who finish the questionnaire receive payments from the game and are paid 96.33 JPY on average.

## Data analysis

With the cross-sectional data of the aforementioned variables, we first characterize generativity in relation to inquisitiveness, and second, characterize happiness in relation to inquisitiveness and generativity along with other factors. We decided to rely on cross-sectional data following some previous researches in that the effectiveness of cross-sectional data analyses is argued for identifying correlation and causal relation among psychometric and sociodemographic variables, especially when the causal direction is somewhat obvious or intuitively straightforward [75, 76, 79]. Specifically, we use mean-based and median regressions to address the two open questions posed in this paper. Question 1: "Does inquisitiveness play a role in generativity?" Question 2: "How does inquisitiveness, along with generativity, affect people's happiness?" To answer questions (1) and (2), regression models are applied to characterize generativity and happiness as dependent variables, respectively, in relation to other key independent variables as described in Fig 1, enabling to identify of important determinants. For empirically characterizing the generativity of subject $i$, the model is specified as

$$\text{generativity}_i = \alpha_0 + \alpha_1 \cdot \text{inquisitiveness}_i + \alpha_2 \cdot \text{SVO}_i + \alpha_3 \cdot \mathbf{x}_i' + \epsilon_i, \tag{1}$$

where $\mathbf{x}_i$ is a vector of sociodemographic independent variables including household income, marital status, family type, education, gender, etc. The associated coefficients of $\alpha_0, \alpha_1, \alpha_2, \boldsymbol{\alpha}_3$ are the parameters to be estimated, and $\epsilon_i$ is a disturbance term. In Eq 1, parameter $\alpha_1$ is of particular interest to statistically examine question (1). For the happiness of subject $i$, the model is

$$\text{SWB}_i = \beta_0 + \beta_1 \cdot \text{inquisitiveness}_i + \beta_2 \cdot \text{generativity}_i + \beta_3 \cdot \text{SVO}_i + \beta_4 \cdot \mathbf{x}_i' + \varepsilon_i \tag{2}$$

where $\text{SWB}_i$ stands for subject $i$'s happiness. The coefficients, $\beta_0, \beta_1, \beta_2, \beta_3, \boldsymbol{\beta}_4$, are parameters to be estimated and $\varepsilon_i$ is a disturbance term. In Eq 2, parameters $\beta_1$ and $\beta_2$ are of particular interest to statistically test question (2).

The median regression is used to statistically analyze the determinants of generativity and happiness in place of parametric mean-based regressions, when observations of generativity

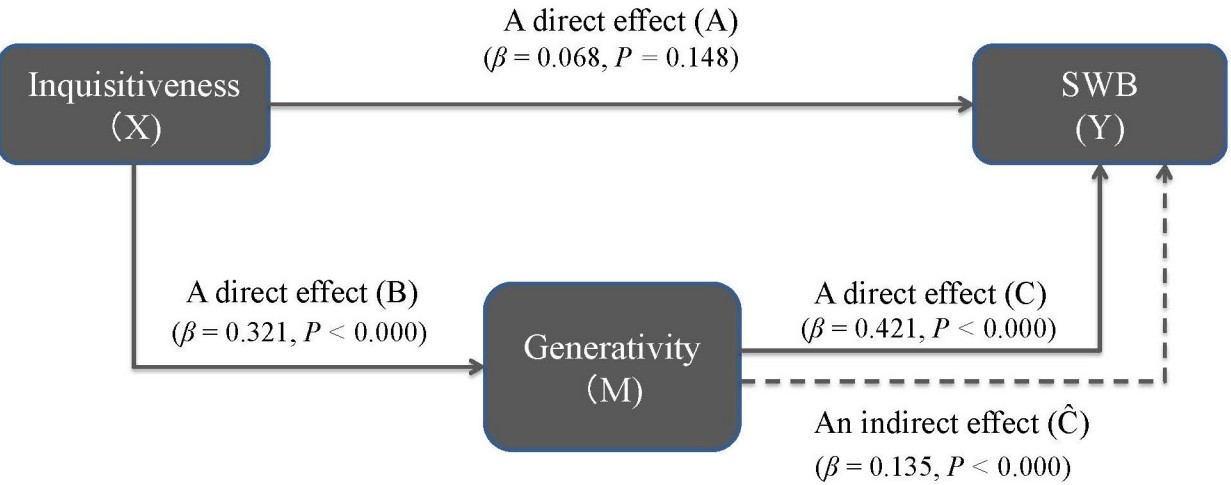

**Fig 3. The mediating effects among inquisitiveness, generativity and SWB.**

and happiness in the sample are considered to be non-normally distributed and/or skewed. The literature claims that median or quantile regressions are more appropriate than parametric mean-based ones, such as ordinary least squares (OLS) regression, yielding robust estimations against the boundary values and/or outliers, especially when the dependent variable is bounded on a certain support range, non-normally distributed and skewed [95, 96]. In fact, we have run Shapiro-Wilk tests for the two dependent variables of generativity and happiness to check their normality with a null hypothesis that the variable is normally distributed. The results do not reject the null hypothesis ($z = 0.630$, $P = 0.264$) for generativity but reject it ($z = 3.621$, $P < 0.01$) for happiness. Therefore, we use the mean-based OLS and median regressions for generativity and happiness with the specifications of Eqs 1 and 2, respectively.

To further confirm the regression results, we apply structural equation modeling (SEM) to analyze whether or not the relationships, i.e., "paths," exist: (1) inquisitiveness → generativity, (2) inquisitiveness → SWB and (3) generativity → SWB. Specifically, the existence of these three paths is examined to establish whether or not generativity is a mediator in the relationship between inquisitiveness and SWB, as graphically conceptualized in Fig 3. To this end, the SEM is one of the effective approaches and enables us to test the paths among the three variables together with the direct and indirect effects of inquisitiveness, following the procedures [97–99]. The SEM analysis computes a beta weight as a standardized coefficient, ($\beta$), along with the associated statistical significance for each path. The magnitudes of standardized coefficients can be directly compared for the purpose of estimating the relationships' relative strength, and the standardization is a necessary step to compare indirect and direct effects among different sets of paths in the same model, i.e., comparison between direct and indirect pathways in a mediation model [100–102].

### Ethics statement

This study was approved by the research ethics committee of Kochi University of Technology. Subjects provided their written consent to participate in this study.

### Results

Tables 1 and 2 present the definitions of all variables used in the analysis and the summary statistics for urban, nonurban and overall areas. The percentage of female subjects and the mean

**Table 1. Variable definitions.**

| Variables | Descriptions |
|---|---|
| Gender | Gender is a dummy variable that takes 1 when the subject is female, otherwise 0. |
| Age | Age is defined as years of age. |
| Marital status | Marital status is a dummy valuable that categorical variable of 0 and 1 where nonmarried (i.e., single, divorce or bereavement) and married are coded as 0 and 1, respectively. |
| Family type | Family type is that categorical variable of 0 and 1 where family type, nuclear family, extended family are coded as 0 and 1 respectively. |
| Area | Area is that categorical variable of 0 and 1 where residential area, nonurban areas, urban areas are coded as 0 and 1, respectively. |
| Education | Education is categorical variables of 1, 2, 3, 4 and 5 where educational background, No scholastic, Junior highschool, Highschool, Undergraduate and Graduate are coded as 1, 2, 3, 4 and 5, respectively. |
| Household income | Household is categorical variables of 1, 2, 3, 4, 5 and 6 where household income per year in JPY, $0 < 1M$, $1 < 2.5M$, $2.5 < 4M$, $4 < 7M$, $7 < 10M$ and more than 10M, respecively. |
| Generativity | Generativity is defined as the measurement of the Loyola generative scale (Range is between 0 and 60). |
| Subjective wellbeing (SWB) | SWB is defined as the measurement of the satisfaction with life scale (SWLS) (Range is between 5 and 35) |
| Inquisitiveness | Inquisitiveness is defined as the measurement by a subscale of the critical thinking disposition scale (Range is between 10 and 50). |
| SVO | The "SVO" represents a dummy valuable taking 1 when the subject is prosocial and otherwise 0, based on SVO games. |

age are similar between urban and nonurban areas (45% and 49% as well as 50.29 years and 49.30 years). Concerning marital status, the percentage of married subjects in urban areas (nonurban areas) is 70% (64%). The percentage of subjects with extended families in urban areas (nonurban areas) is 11% (20%). Subjects in urban and nonurban areas possess a college degree and a high school diploma as the median education level, respectively. The median household income in urban areas is the same as that in nonurban areas. Contrary to our expectations, nonurban areas have a slightly higher percentage of unmarried individuals than do urban areas in our survey. This suggests that currently, in Japan, urban and nonurban people's lives are similar except regarding family type. Table 2 shows the summary statistics of subjects' generativity in urban, nonurban and overall areas. We have computed Cronbach's alpha for this scale to be 0.90, illustrating that the generativity scale possesses acceptable internal consistency in our sample. The median generativity score is 26 points in both urban and nonurban areas, while the average generativity scores are 25.87 and 24.63 points, respectively. This finding suggests that generativity between urban and nonurban subjects is similar; however, mean generativity in urban subjects is slightly higher than that in nonurban subjects.

Table 2 shows the summary statistics of subjective wellbeing (see the "SWB" row in Table 2) in urban, nonurban and overall areas. We have computed Cronbach's alpha for this scale to be 0.93, illustrating that the satisfaction with life scale (SWLS) possesses acceptable internal consistency in our sample. The median scores of the SWLS are 19 and 18 points in urban and nonurban areas, while the average scores of the SWLS are 17.82 and 17.53 points, respectively. This finding suggests that SWB between urban and nonurban subjects is not distinct. Table 2 also shows the summary statistics of subjects' inquisitiveness in urban, nonurban and overall areas. We have computed Cronbach's alpha for this scale to be 0.94, illustrating that the inquisitiveness scale possesses acceptable internal consistency in our sample. The median score of inquisitiveness is 32 points in both urban and nonurban areas, while the average scores of inquisitiveness are 32.20 and 32.30 points, respectively. This finding suggests that inquisitiveness between urban and nonurban subjects is not different.

**Table 2. Summary statistics of subject's sociodemographic information and major variables.**

| Variables | Urban areas | | | | | Rural areas | | | | | Overall areas | | | | |
|---|---|---|---|---|---|---|---|---|---|---|---|---|---|---|---|
| | **M** | **Me** | **SD** | **Min** | **Max** | **M** | **Me** | **SD** | **Min** | **Max** | **M** | **Me** | **SD** | **Min** | **Max** |
| Gender (female) | 0.45 | 0 | 0.50 | 0 | 1 | 0.49 | 0 | 0.50 | 0 | 1 | 0.47 | 0 | 0.50 | 0 | 1 |
| Age | 50.29 | 51 | 17.40 | 20 | 88 | 49.30 | 49 | 16.10 | 20 | 88 | 49.79 | 50 | 16.74 | 20 | 88 |
| Marital status (experienced) | 0.70 | 1 | 0.46 | 0 | 1 | 0.64 | 1 | 0.48 | 0 | 1 | 0.67 | 1 | 0.47 | 0 | 1 |
| Family type (extended) | 0.11 | 0 | 0.31 | 0 | 1 | 0.20 | 0 | 0.40 | 0 | 1 | 0.15 | 0 | 0.36 | 0 | 1 |
| Education | 3.73 | 4 | 0.58 | 1 | 5 | 3.46 | 3 | 0.64 | 1 | 5 | 3.61 | 4 | 0.62 | 1 | 5 |
| Household income | 3.86 | 4 | 1.40 | 1 | 6 | 3.59 | 4 | 1.33 | 1 | 6 | 3.72 | 4 | 1.37 | 1 | 6 |
| Generativity | 25.87 | 26 | 10.33 | 3 | 51 | 24.63 | 26 | 9.38 | 2 | 47 | 25.25 | 26 | 9.87 | 2 | 51 |
| SWB | 17.82 | 19 | 6.84 | 5 | 35 | 17.53 | 18 | 6.46 | 5 | 33 | 17.67 | 19 | 6.65 | 5 | 35 |
| Inquisitiveness | 32.20 | 32 | 7.39 | 10 | 50 | 32.30 | 32 | 7.23 | 10 | 50 | 32.25 | 32 | 7.30 | 10 | 50 |
| SVO (Prosocial) | 0.62 | 1 | 0.49 | 0 | 1 | 0.64 | 1 | 0.48 | 0 | 1 | 0.63 | 1 | 0.48 | 0 | 1 |
| Subjects | *n = 200* | | | | | *n = 200* | | | | | *n = 400* | | | | |

SD stands for standard deviation.

Next, we report the summary statistics of subjects' SVOs, focusing on the percentages of prosocial subjects in urban, nonurban and overall areas (see the last row of "SVO (prosocial)" in Table 2). While 63% of subjects in the overall are prosocial, 62% (64%) of urban (nonurban) subjects are prosocial. This result is in sharp contrast with similar studies in developing countries showing that the percentages of prosocial subjects between urban and rural areas are quite different, and the percentage of prosocial subjects in rural areas is higher than that in urban ones [15, 80, 81]. This finding suggests that the degree of prosociality among people is similar between urban and nonurban areas in Japan, compared to other developing countries.

To empirically characterize open question (1), we perform ordinary least squares (OLS) regression in which generativity is taken as a dependent variable, and inquisitiveness is taken as an independent one along with other factors, as described in Eq 1. Table 3 reports the estimated coefficients ($\alpha_1, \alpha_2, \boldsymbol{\alpha}_3$) and their respective standard errors of the independent variables on generativity, along with statistical significance. Model 1 in Table 3 contains inquisitiveness and age as independent variables. Next, we gradually add marital status, the gender dummy and other factors as independent variables in models 2 to 4, building upon model 1. We first find that inquisitiveness is statistically significant with a positive sign at 1% in a robust manner, irrespective of the models. The estimated coefficients of inquisitiveness on subjects' generativity range between 0.390 and 0.395 in models 1 to 4, implying that a subject is likely to have an increase in generativity by the range, when one unit in her inquisitiveness rises.

Second, age has a positive effect on the subject's generativity at 1% significance in models 1 to 4. The estimated coefficients of age in models 1 to 4 indicate that a subject is likely to increase generativity by 0.086 ∼ 0.110 when she ages by one year. Marital status also exhibits 1% and 5% statistical significance with a positive sign in models 2 to 4, implying that a married subject tends to enhance her generativity by 2.259 ∼ 2.471, as compared with a nonmarried subject. The other independent variables, such as gender, prosociality, education, household income and area, are identified as statistically insignificant, as shown in models 2 to 4 in Table 3. We confirm that the main results qualitatively remain the same, irrespective of the various specifications of models other than models 1 to 4, such as the inclusion of age squared and/or interaction terms among the variables. Overall, inquisitiveness, age and marital status are confirmed to be the main determinants of subjects' generativity.

**Table 3. Estimation results of OLS regression on people's generativity.**

| Variables | Generativity | | | |
|---|---|---|---|---|
| | **Model 1** | **Model 2** | **Model 3** | **Model 4** |
| Inquisitiveness | 0.395*** | 0.390*** | 0.391*** | 0.391*** |
| | (0.064) | (0.063) | (0.064) | (0.064) |
| Age | 0.110*** | 0.086*** | 0.088*** | 0.090*** |
| | (0.028) | (0.029) | (0.029) | (0.030) |
| Marital status (base group = non married) | | 2.458*** | 2.471*** | 2.259** |
| | | (0.978) | (0.984) | (1.047) |
| Gender (base group = male) | | | −0.632 | −0.570 |
| | | | (0.923) | (0.936) |
| Prosociality (base group = proself) | | | −0.479 | −0.463 |
| | | | (0.952) | (0.954) |
| Education | | | | −0.029 |
| | | | | (0.744) |
| Household income | | | | 0.147 |
| | | | | (0.360) |
| Area (base group = nonurban) | | | | 0.950 |
| | | | | (0.939) |

***significant at 1 percent,

**significant at 5 percent,

*significant at 10 percent

To empirically characterize open question (2), we perform the median regression in which SWB is taken as a dependent variable, and generativity and inquisitiveness are taken as an independent one along with other factors, as described in Eq 2. Table 4 reports the estimated coefficients ($\beta_1$, $\beta_2$, $\beta_3$, $\boldsymbol{\beta}_4$) and their respective standard errors of the independent variables on SWB, along with statistical significance. Model 1 of Table 4 contains generativity and inquisitiveness as independent variables, and next, we gradually add marital status, age, household income and other factors as independent variables in models 2 to 4, building upon model 1. We first find that the generativity is statistically significant with a positive sign at 1% in a robust manner, irrespective of the models. The estimated coefficients of generativity on subjects' SWB range between 0.265 and 0.293 in models 1 to 4, implying that a subject is likely to increase her SWB by the range when one unit in her generativity rises.

Second, inquisitiveness has a positive effect on people's SWB at 5% and 10% significance in models 1 and 4. The estimated coefficients of inquisitiveness in models 1 to 4 suggest that a subject is likely to increase her SWB range between 0.083 and 0.108 when one unit in her inquisitiveness rises. Marital status also exhibits 1% and 5% statistical significance with a positive sign in models 2 to 4, implying that a married subject tends to enhance her SWB by 1.773 ∼ 2.311, as compared with a nonmarried subject. Similarly, in models 2 to 4, a subject is likely to enhance her SWB range by 0.045 ∼ 0.052 at 5% significance when she ages by one year. The other independent variables, such as household income, gender, prosociality, education, family type and area, are identified to be statistically insignificant, as shown in models 3 to 4 in Table 4. We confirm that the main results qualitatively remain the same, irrespective of the various specifications of models other than models 1 to 4, such as age squared or interaction terms among the variables.

**Table 4. Estimation results of median regression on subjective wellbeing (SWB).**

| Variables | SWB | | | |
|---|---|---|---|---|
| | **Model 1** | **Model 2** | **Model 3** | **Model 4** |
| Generativity | 0.293*** | 0.267*** | 0.269*** | 0.265*** |
| | (0.042) | (0.039) | (0.039) | (0.040) |
| Inquisitiveness | 0.108** | 0.083* | 0.083* | 0.098* |
| | (0.057) | (0.052) | (0.051) | (0.053) |
| Marital status (base group = non married) | | 2.311*** | 1.773** | 1.784** |
| | | (0.771) | (0.801) | (0.842) |
| Age | | 0.045** | 0.047** | 0.052** |
| | | (0.023) | (0.023) | (0.024) |
| Household income | | | 0.285 | 0.325 |
| | | | (0.272) | (0.289) |
| Gender (base group = male) | | | 0.284 | 0.297 |
| | | | (0.710) | (0.621) |
| Prosociality (base group = proself) | | | −0.311 | −0.252 |
| | | | (0.730) | (0.765) |
| Education | | | | 0.297 |
| | | | | (0.621) |
| Family type (base group = nuclear family) | | | | −0.741 |
| | | | | (1.036) |
| Area (base group = nonurban) | | | | −0.385 |
| | | | | (0.756) |

1 ***significant at 1 percent, **significant at 5 percent, *significant at 10 percent

2 We have run median regression including independent variables of age squared and household income squared. The result shows less influence from independent variables of them on subjective wellbeing. Based on the outcome, we judge that these variables could be removed from the models to simplify showing regression results.

We use the SEM analysis to check the regression results as another confirmation for the existence of the relationship within key variables. We first analyze the two direct effects from inquisitiveness to SWB (path $A$ in Fig 3) and from generativity to SWB (path $C$ in Fig 3). The results demonstrate the existence of path $A$ ($\beta = 0.068$, $p = 0.148$) and that of path $C$ ($\beta = 0.421$, $p < 0.000$), meaning that both inquisitiveness and generativity appear to have some direct effects on SWB. Next, we analyze the direct effect from inquisitiveness to generativity (path $B$ in Fig 3) and an indirect effect from inquisitiveness to SWB through generativity (path $\hat{C}$ in Fig 3). The analyses demonstrate the significance of path $B$ ($\beta = 0.321$, $p < 0.000$) as well as that of path $\hat{C}$ ($\beta = 0.135$, $p < 0.000$). Comparing direct vs. indirect paths from inquisitiveness to SWB in the mediation model, the magnitude of path $\hat{C}$ ($\beta = 0.135$, $p < 0.000$) is found to be stronger than that of path $A$ ($\beta = 0.068$, $p = 0.148$). These results show that the indirect path $\hat{C}$ from inquisitiveness to SWB plays a crucial role through a mediator of generativity, gaining consistent results with the regression results. Overall, generativity and inquisitiveness are confirmed as the main determinants for characterizing subjects' SWB.

## Discussion

We are now ready to summarize the answers to the two open questions posed at the end of the introduction section. As described in our conceptual framework of Fig 1, it is well known that happiness is mainly characterized by the three factors, such as cognitive factors, noncognitive factors and sociodemographic factors. The first question is, "Does inquisitiveness play a role in

generativity?" Our answer to this question is that inquisitiveness, ($\alpha_1$), is the crucial determinant regarding whether people possess high generativity in Fig 1. Inquisitiveness is of utmost importance due to regression and SEM analyses' magnitude and statistical significance. The second question is, "How does inquisitiveness along with generativity affect people's happiness?" Our answer to this question is that generativity, ($\beta_2$), and inquisitiveness, ($\beta_1$), directly and indirectly, affect subjective happiness, demonstrating the importance of possessing inquisitiveness and generativity for SWB in Fig 1.

Some studies have pointed out that inquisitiveness is stable as a part of critical thinking disposition, even in the long run, and considered innate because even very young children actively ask adults many questions and pursue explanatory information due to their curiosity [71, 73, 74, 103–105]. Conversely, other studies have pointed out that inquisitiveness can be acquired and further enhanced by learning [6, 7, 59, 103, 106–108]. For instance, Sannomiya and Yamaguchi [109] conduct an experiment with 100 Japanese junior high school students, establishing that inquisitiveness and critical thinking ability are fostered with training and meta-cognitive belief. In addition, some leadership training programs have been developed to enhance inquisitiveness in business because an inquisitive person is considered able to improve productivity, creativity and management in practice [60, 61, 66, 70, 71].

Based on the above discussions, inquisitiveness can plausibly be considered to increase through education, experiences and training, i.e., as a part of culture, in the course of people's lifetimes. If this is true, then the analyses in this paper suggest that both generativity and happiness are expected to increase, as people become inquisitive through such cultural activities, i.e., education, experience and training. It is argued that subjective wellbeing has a positive correlation with the achievement of sustainable development goals (SDGs) [110–112]. At the same time, generativity is known to contribute to SDGs, because it facilitates intergenerational cultural and resource transfers between current and future generations [15, 55, 81]. With these findings in mind, an important contribution of this study that it provides statistical evidence that inquisitiveness is a fundamental human attribute to enhance not only generativity but also people's happiness, possibly leading to the materialization of sustainable societies.

We note some limitations of our research and directions for future research. It should be noted that Japanese cross-sectional data are collected, utilized and analyzed in this study, excluding very young people. Further research shall be conducted to confirm the robustness of our results by spanning such young people or by considering different societies, such as western or other Asian countries, providing some insight about age and cultural differences. At this point in time, we conjecture that inquisitiveness remains consistent and important even in different ages and countries, being in line with our study. Moreover, as some studies have argued, it shall be desirable to collect and examine panel data to confirm and generalize our findings [113–115]. To this end, experimental methods can be employed to collect panel data and examine the possible causality among inquisitiveness, generativity and happiness. These caveats notwithstanding, it is our belief that this research is an essential first step toward understanding the importance of inquisitiveness along with generativity and happiness, hoping that further studies will ensure to identify how to enhance people's happiness and sustainability of societies.

## Conclusions

This paper addresses how generativity and happiness are characterized by inquisitiveness. We hypothesize that inquisitiveness is an essential determinant for generativity and happiness, empirically examining the relationships along with sociodemographic, cognitive and noncognitive factors. To this end, we conduct questionnaire surveys with 400 Japanese subjects to

collect sociodemographic, cognitive and noncognitive factors, applying the analysis of OLS regression, median regression and structural equation modeling (SEM). First, the analyses identify the importance of inquisitiveness in characterizing generativity in that inquisitive people tend to be generative. Second, people are identified to be happy as they have high inquisitiveness and generativity, demonstrating two influential roles of inquisitiveness, directly and indirectly, through a mediator of generativity. Overall, the results suggest that inquisitiveness (curiosity & acceptance of something and someone different and/or new) is a main engine for one person to enhance generativity and happiness through intergenerational and intragenerational communication or relations.

## Supporting information

**S1 Data. Excel "Inquisitiveness.xlsx" data file.** It contains all the necessary data to replicate the statistical and regression results presented in this paper.
(XLSX)

## Acknowledgments

The authors thank anonymous referees, Zhang Jingchao, Makoto Kakinaka, Hiroaki Miyamoto, Mostafa Shahen, Kenta Tanaka and Raja Rajendra Timilsina for their helpful advice, comments and supports.

## Author Contributions

**Supervision:** Koji Kotani.

**Visualization:** Junichi Hirose.

**Writing – original draft:** Junichi Hirose.

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
