## [Decision Letter · Decision Letter 0]

23 Nov 2021

PONE-D-21-26698How does inquisitiveness matter for generativity and happiness?PLOS ONE

Dear Dr. Kotani,

Thank you for submitting your manuscript to PLOS ONE. After careful consideration, we feel that it has merit but does not fully meet PLOS ONE’s publication criteria as it currently stands. Therefore, we invite you to submit a revised version of the manuscript that addresses the points raised during the review process.

Two Reviewers well assessed this manuscript. Minor revisions are necessary in the present form. See the Reviewers' comments and respond them appropriately.

We look forward to receiving your revised manuscript.

Kind regards,

Masaki Mogi

Academic Editor

PLOS ONE

Journal Requirements:

2. Please modify the title to ensure that it is meeting PLOS’ guidelines (https://journals.plos.org/plosone/s/submission-guidelines#loc-title). In particular, the title should be "specific, descriptive, concise, and comprehensible to readers outside the field" and in this case it is not informative and specific about your study's scope and methodology.

We are also grateful to the financial support from Kochi Uni433 versity and Kochi University of Technology

The authors received no specific funding for this work.

Reviewers' comments:

Reviewer's Responses to Questions

**Comments to the Author**

1. Is the manuscript technically sound, and do the data support the conclusions?

Reviewer #1: Yes

Reviewer #2: Yes

2. Has the statistical analysis been performed appropriately and rigorously? 

Reviewer #1: Yes

Reviewer #2: Yes

3. Have the authors made all data underlying the findings in their manuscript fully available?

Reviewer #1: No

Reviewer #2: Yes

4. Is the manuscript presented in an intelligible fashion and written in standard English?

Reviewer #1: Yes

Reviewer #2: No

5. Review Comments to the Author

Reviewer #1: In this manuscript, the Authors aimed to How does inquisitiveness matter for generativity and happiness?. The manuscript is well designed and written.

1- I would add a short comment in the Discussion section to analyse if the data collected are in line with similar investigations from western and other Asian countries . This would provide an idea about cultural differences, psychological issues, and lifestyle differences on inquisitiveness.

2- I suggest shortening the introduction

Reviewer #2: 1. In general, the Introduction is adequate, providing background, motivation, definitions and a fair overview of studies in the areas addressed by the study. Two open questions are provided explicitly.

2. The reported methods are sound.

3. The data and the results support the conclusions.

Figure 4, illustrating the SEM model, should include usual statistical results, for better and faster reference of the reader.

4. The manuscript would benefit from a better organization.

(Related to 4. Is the manuscript presented in an intelligible fashion and written in standard English? Answer: No)

Suggestion:

. Introduce subsections in 2. Materials and methods, e.g., Participants and procedures, Measures, Data analysis, including information about SEM that is now misplaced in Results.

. Create a Discussion section, including a discussion of own results, comparison with results from others, contributions and maybe limitations and directions for future work.

. Conclusions would then restate objective and hypothesis (already there) and bear a short summary of methods and main results.

5. The standard of English is good. The following statement requires attention. SEM’s results are a proof, not a cause, like the text seems to imply.

“Inquisitiveness is of utmost importance due to regression and SEM analyses’ magnitude and statistical significance.”

6. Providing the results of adequate tests, maybe you could skip Figure 3a and Figure 3b.

6. PLOS authors have the option to publish the peer review history of their article (what does this mean?). If published, this will include your full peer review and any attached files.

Reviewer #1: No

Reviewer #2: No

---

## [Author Response · Author response to Decision Letter 0]

28 Dec 2021

Reviewer1: Thank you for spending your time on our manuscript and providing us with your comments. Your comments are very constructive and helpful for the improvement of our paper. We have tried to clarify all of your quarries and modified our manuscript accordingly. Our responses to your comments are given in the attached document.

Reviewer2: Thank you for spending your time on our manuscript and providing us with your comments. Your comments are very constructive and helpful for the improvement of our paper. We have tried to clarify all of your quarries and modified our manuscript accordingly. Our responses to your comments are given in the attached document.

---

## [Decision Letter · Decision Letter 1]

7 Feb 2022

How does inquisitiveness matter for generativity and happiness?

PONE-D-21-26698R1

Dear Dr. Kotani,

We’re pleased to inform you that your manuscript has been judged scientifically suitable for publication and will be formally accepted for publication once it meets all outstanding technical requirements.

Kind regards,

Masaki Mogi

Academic Editor

PLOS ONE

Additional Editor Comments (optional):

No further comment.

Reviewers' comments:

Reviewer's Responses to Questions

**Comments to the Author**

1. If the authors have adequately addressed your comments raised in a previous round of review and you feel that this manuscript is now acceptable for publication, you may indicate that here to bypass the “Comments to the Author” section, enter your conflict of interest statement in the “Confidential to Editor” section, and submit your "Accept" recommendation.

Reviewer #1: (No Response)

Reviewer #2: (No Response)

2. Is the manuscript technically sound, and do the data support the conclusions?

Reviewer #1: Yes

Reviewer #2: Yes

3. Has the statistical analysis been performed appropriately and rigorously? 

Reviewer #1: Yes

Reviewer #2: Yes

4. Have the authors made all data underlying the findings in their manuscript fully available?

Reviewer #1: Yes

Reviewer #2: Yes

5. Is the manuscript presented in an intelligible fashion and written in standard English?

Reviewer #1: Yes

Reviewer #2: Yes

6. Review Comments to the Author

Reviewer #1: (No Response)

Reviewer #2: Anyway, as stated before, I have accepted the paper without further input to the authors.

They have answered my questions and chose to follow the suggestions.

I believe the quality of the paper has improved.

7. PLOS authors have the option to publish the peer review history of their article (what does this mean?). If published, this will include your full peer review and any attached files.

Reviewer #1: No

Reviewer #2: No

---

## [Editor Report · Acceptance letter]

11 Feb 2022

PONE-D-21-26698R1 

How does inquisitiveness matter for generativity and happiness? 

Dear Dr. Kotani:

I'm pleased to inform you that your manuscript has been deemed suitable for publication in PLOS ONE. Congratulations! Your manuscript is now with our production department. 

Kind regards, 

on behalf of

Dr. Masaki Mogi 

Academic Editor

PLOS ONE